# Antagonism of *Bacillus velezensis* Isolate from Anaerobically Digested Dairy Slurry against Fusarium Wilt of Spinach

**Tomomi Sugiyama [1], Keiko T. Natsuaki [2], Naoto Tanaka [3], Yuh Shiwa [3] and Mami Irie [2],***

[1] NARO Institute of Vegetable and Floriculture Science, Tsukuba 305-8519, Japan; tomomi.s@affrc.go.jp

[2] Department of International Agricultural Development, Faculty of International Agriculture and Food Studies, Setagaya Campus, Tokyo University of Agriculture, Tokyo 156-8502, Japan; keiko@nodai.ac.jp

[3] Department of Molecular Microbiology, Faculty of Life Sciences, Setagaya Campus, Tokyo University of Agriculture, Tokyo 156-8502, Japan; n3tanaka@nodai.ac.jp (N.T.); y3shiwa@nodai.ac.jp (Y.S.)

* Correspondence: mami-o@nodai.ac.jp; Tel.: +81-3-5477-2489

**Abstract:** This study was designed to assess the suppressive effects of various anaerobically digested slurries (ADSs), and the microorganisms inhabiting them, against Fusarium wilt in spinach. We used five different ADSs from a range of source materials (dairy cow manure, sewage sludge, food garbage, pig manure, night soil sludge), combined in different proportions. All five raw ADSs suppressed the growth of *Fusarium oxysporum* f. sp. *spinaciae* (Fos) on agar plates using a co-culture test. In contrast, filtrate ADSs did not suppress the growth of Fos. In total, 32 bacterial strains were isolated from five ADSs, and eight isolates showed antagonistic activities against Fos. Based on 16S rDNA sequences, the strain AD-3 isolated from ADS from dairy cow manure belonged to *Bacillus velezensis*. Genome analysis revealed that AD-3 had two kinds of genes related to the production of the non-ribosomal lipopeptides, fengycin/plipastatin (*pps* genes), and surfactin (*srf* genes). In pot assays, inoculation of AD-3 ($1.0 \times 10^6$ CFU·g$^{-1}$ dry soil) into Fos-infected soil ($1.0 \times 10^5$ bud-cells·g$^{-1}$ dry soil) significantly reduced the severity of Fusarium wilt disease at 28 d after seedling. The percentage reductions in disease severity in two replicates were 64.3% and 44.3%, respectively. Thus, bacterial strain AD-3 could be applied to reduce Fusarium wilt in spinach.

**Keywords:** biological control; anaerobically digested dairy slurry; *Bacillus velezensis*

## 1. Introduction

Fusarium wilt disease, caused by *Fusarium oxysporum* f. sp. *spinaciae* (Fos), is a serious soil-borne disease. *Fusarium oxysporum* has high host specificity and is responsible for severe damage to economically important plants [1]. *F. oxysporum* has been positioned within the top 10 of economically significant plant-pathogenic fungi [2]. Chemical fungicides or soil fumigation are commonly used to control the disease. However, the use of fumigants such as methyl bromide has been banned in many parts of the world. Application of chemical products negatively affects human health, function of ecosystems, and microorganisms in soil [3,4]. The development of alternatives to these conventional controls is urgently needed for sustainable agricultural practices.

Organic amendments can be used to improve soil quality and manage soil-borne diseases [5]. Among available organic amendments, compost is the most commonly studied. Composting is a common method of recycling organic waste, and the final by-product can be used in agriculture [6]. Suppressive effects of compost against soil-borne fungal diseases have been reported [7,8]. Anaerobic waste treatment is another solution for treatment of agro-industrial waste and the organic parts of source-separated household waste. The treatment produces biogas as a source of renewable energy and anaerobically digested slurry (ADS) as a liquid residue [9,10]. Scientific reports of anaerobic treatment, especially biogas, have increased rapidly in terms of renewable energy technology [11]. In addition, research on ADS has continuously increased since 2007–2009 [11]. In general,

ADS has a high concentration of nutrients such as nitrogen, phosphorus, and potassium, which are available in a suitable form for plants to absorb, meaning that ADS can be used as liquid fertilizer. When considering ADS for agronomic use, the risks for crop production should be assessed because the chemical composition of an ADS depends on the primary source materials [12,13]. In Europe, the application of ADS is recommended based on specific regulations and guidelines [14,15]. Some studies reported that ADS is a valuable alternative to fertilizer in agricultural production [16,17].

The use of ADS has mainly focused on functions such as nutrient availability, crop productivity, and reusing organic waste [18–20]. The role of ADS in plant disease control has been studied. For example, ADS sourced from pig manure suppressed the growth of plant pathogenic fungi in vitro [21] and the application of ADS to soil has been shown to suppress several plant diseases including *Ralstonia* spp. [22], *Fusarium* spp. [22], and *Phytophthora* spp. [23]. These reports focused on physicochemical properties as factors for plant disease control. However, microorganisms in ADS may also have a suppressive effect. The disease control ability of organic amendments and ADS varies depending on the source materials. The use of ADS for plant disease control has not been well studied compared with other types of organic amendments. Evaluation of the effects of ADS on disease suppression can improve our understanding of its potential applications. The first objective of this study was to assess the efficacy of ADS generated from different source materials in suppressing *Fusarium oxysporum* f. sp. *spinaciae* (Fos). We isolated bacteria from ADS and tested the effects of these against Fos in vitro. The second objective was to verify the antagonistic effect of a selected bacterial isolate against Fusarium wilt of spinach in vivo.

## 2. Materials and Methods

### 2.1. Sampling and Characteristics of ADSs

In this study, five ADSs were generated from different source materials: dairy cow manure (AD), sewage sludge (AS), food garbage (AF), pig manure + food garbage (APF), and sewage sludge + night soil sludge + food garbage (ASNF). Mesophilic fermentation of anaerobic biological treatment in facilities occurred at 35 °C. All digestates were taken from a methane fermentation tank running in continuous mode without dewatering, and the slurry was used for experiments. After sampling, the slurry was stored in 20 L plastic tanks at 4 °C. The details of each ADS are presented in Table 1.

**Table 1.** Source material composition, processing conditions, and facility location for five anaerobically digested slurries (ADSs).

| Sample [1] | Rate of Source Materials | | | | | Processing Condition | | Microbial Immobilization Method | Pre-Treatment | Facility Location |
|---|---|---|---|---|---|---|---|---|---|---|
| | Dairy Cow Manure | Sewage Sludge | Food Garbage | Pig Manure | Night Soil Sludge | Treatment Temperature | HRT [2] | | | |
| | (%) | (%) | (%) | (%) | (%) | (°C) | (Days) | | | |
| AD | 100 | | | | | 35–36 | 45 | Anaerobic fluidized bed process | Solid liquid separation | Tochigi prefecture |
| AS | | 100 | | | | 35–36 | 30 | Anaerobic contact process | Gravitation enrichment | Yamagata prefecture |
| AF | | | 100 | | | 35–36 | 20 | Anaerobic fluidized bed process | Acid fermentation | Hokkaido |
| APF | | | 10 | 90 | | 35–36 | 30 | Anaerobic fluidized bed process | Crush | Okayama prefecture |
| ASNF | | 74 | 9 | | 17 | 35–36 | 30 | Anaerobic fluidized bed process | Acid fermentation | Fukuoka prefecture |

[1] ADSs generated from different source materials. AD, dairy cow manure; AS, sewage sludge; AF, food garbage; APF, pig manure + food garbage; ASNF, sewage sludge + night soil sludge + food garbage. [2] HRT, Hydraulic Retention Time.

### 2.2. Antifungal Activity of Raw and Filtrate ADSs

The plant pathogen *Fusarium oxysporum* f. sp. *spinaciae* (Fos, MAFF: 103059) was used. Fos was cultured on potato-dextrose agar (PDA) (Nihon pharmaceutical Co., Ltd., Tokyo,

Japan) in petri dishes ($\varphi$ 90 × 15 mm) at 25 °C for 14 d and used as inocula. Raw and filtrate ADSs were used for the assay. Filtrate ADSs were prepared as follows: 10 mL of each raw ADS was centrifuged at 3000 rpm for 20 min, and the supernatant was filtered through a 0.2 μm disposable membrane filter (Toyo Roshi Kaisha, Ltd., Tokyo, Japan). The fungal colony was removed as a small colony disk using a 5 mm cork borer, and one side was placed on a PDA plate. Sterilized filter paper disks ($\varphi$ 5 mm) were placed in 10 mL of each raw and filtered ADS for 10 min to absorb it, and then placed on the other side of the PDA plates. A pathogen-only plate was used as a control. All treatments had five replicates. All plates were incubated for 18 d at 25 °C in the dark. After incubation, two plates were randomly selected from each treatment. Photos were taken of the plates and the digital images were used to measure the diameter of any colonies developed from the mycelial disk, using ImageJ software (version 1.52, NIH, Bethesda, MA, USA). The experiment was repeated twice.

### 2.3. Isolation of Bacteria from Five ADSs

The five ADSs were serially diluted with sterilized water and the dilutions were spread onto nutrient agar (NA, Nissui Pharmaceutical Co., Ltd., Tokyo, Japan). After incubation for 2 to 4 d at 25 °C in the dark, bacterial colonies that appeared on the plates were transferred to NA plates, and single colony isolates were obtained. These isolates were preserved in solution (10 g of skim milk and 1.5 g sodium L-glutamate monohydrate 100 mL$^{-1}$ distilled water) at −20 °C until use.

### 2.4. Antifungal Activity of Bacterial Isolates from ADSs

Bacterial isolates from the five ADSs described in Section 2.3 were used for confrontation assays. Small bacterial colony disks were removed from cultures on NA plates after 24–48 h using a 5 mm cork borer and were placed on one side on PDA plates. Fos fungal colonies were also removed as a small colony disk in the same manner. A pathogen-only plate was used as a control. All treatments had five replicates. All plates were incubated for 18 d at 25 °C in the dark. After incubation, two plates were randomly selected from each treatment. Photographs were taken and the digital images were used to measure the diameter of mycelial colonies that developed from a mycelial disk, using ImageJ software (version 1.52, NIH, Bethesda, MD, USA).

### 2.5. Pot Experiment

2.5.1. Preparation of Fos Inoculum

Fos inoculum was prepared on potato–sucrose broth (200 g potato and 20 g sucrose L$^{-1}$ distilled water) at 25 °C by shaking the culture at 110 rpm for 5 d. The resulting spore suspension was filtered through double gauze to remove the hyphae and was then transferred into sterile 50 mL plastic tubes, centrifuged at 2000 rpm for 3 min, and then the supernatants were discarded. The Fos inoculum was prepared with distilled water and adjusted to $1.0 \times 10^6$ to $1.0 \times 10^7$ bud-cells·mL$^{-1}$ using a hemocytometer (AS ONE Corp., Osaka, Japan).

2.5.2. Preparation of AD-3

In Section 2.4, AD had several effective bacteria against Fos among ADSs. Of the several bacteria in AD, AD-3, which suppressed Fos growth the most, was used for pot experiments. AD-3 cells were grown on NA plates for 24 h, and the culture was grown on nutrient broth media (5 g meat extract, 5 g NaCl, and 10 g peptone L$^{-1}$ distilled water) at 25 °C on a constant rotary shaker at 100 rpm for 48 h. The suspension was transferred into sterile 50 mL plastic tubes, centrifuged at 3000 rpm for 10 min, and the supernatants were discarded (repeated twice). The AD-3 cell pellet was dissolved in sterilized distilled water. The suspension of AD-3 was adjusted to an optical density of 1.0 at 600 nm using a spectrophotometer (Thermo Electron Corp., Waltham, MA, USA).

### 2.5.3. Pot Assay

The experiment was conducted in pots (10.5 × 9 cm) containing the equivalent of 200 g of dried black loam soil. The soil was sieved through 2 mm mesh and soil pH was adjusted to within the range 6.9 to 7.1 using hydrate lime. The amount of hydrate lime was calculated using the Arrhenius equation. The final rates of N, $P_2O_5$, and $K_2O$ were adjusted to 80 kg·ha$^{-1}$ using ammonium sulfate, fused phosphate, and potassium chloride. The pot experiment had three treatments: F, only Fos-inoculated soil; F+AD-3, soil amended with Fos and AD-3; and unamended, non-pathogen control. To prepare Fos-infected soil, the bud-cell suspension of Fos was inoculated into the soil to give a final concentration of $1.0 × 10^5$ bud-cells·g$^{-1}$ dry soil, and pots were incubated for 5 d at 25 °C in the dark. AD-3 suspension was added to the infected soil to give a final concentration of $1.0 × 10^6$ CFU·g$^{-1}$ dry soil. All pots were incubated for 10 d at 25 °C in the dark. During the incubation period, soil moisture was maintained at 60% water-holding capacity (WHC) by spraying with distilled water. As for spinach cultivar, 'Okame' (*Spinacia oleracea* L.; Takii Seed, Kyoto, Japan) with high susceptibility to Fos [24] was used. After incubation, spinach seeds were sown in each pot and the plants were grown in an incubator (day/night: 25/22 °C, 12/12 h). All pots were irrigated daily to keep the soil moisture at 60% WHC.

A total of 60 pots were prepared; 30 pots (15 pots of F and 15 pots of F+AD-3) were used to estimate the density of Fos in the soil. Soil was sampled from three pots that were randomly selected per treatment at 0, 7, 14, 21, and 28 d after seedling, and the density of Fos was measured using Komada selective medium [25]. The other 30 pots (10 pots per treatment) were used to evaluate disease severity at 28 d after seedling. The disease symptoms were categorized using a disease index of 0, no infection; 1, 1–30% of leaves wilted; 2, 31–60% of leaves wilted; 3, 61–90% of leaves wilted; 4, >90% of leaves wilted or dead. Disease severity (DS) was calculated according to the following formula:

$$DS = \sum nd \times 100/4T \tag{1}$$

where $n$ = number of plants in each disease index, $d$ = disease index, and $T$ = total number of plants by treatments. The inhibitory effect (IE) of AD-3, as a percentage reduction in disease severity, was calculated according to the following formula:

$$IE\ (\%) = (100 - DS_{F+AD-3})/DS_F \times 100 \tag{2}$$

where $DS_{F+AD-3}$ = disease severity of F+AD-3, $DS_F$ = disease severity of F. The pot experiments were repeated twice (Experiment 1, Exp 1; Experiment 2, Exp 2). In total, disease severity for 27 plants ($T = 9$) in Exp 1 and 21 plants ($T = 7$) in Exp 2 was investigated.

Statistical analyses were conducted in R v. 3.5.3 software. The antagonistic activity of ADSs and bacterial isolates against Fos in vitro was analyzed with Tukey's HSD test following one-way analysis of variance (ANOVA). The effect of inoculation of AD-3 on Fos density was analyzed using a *t*-test. The data for disease severity were arcsine transformed in advance, and Tukey's HSD test following one-way analysis of variance (ANOVA) was performed to reveal the efficacy of AD-3 against Fusarium wilt of spinach ($p < 0.05$).

### 2.6. Genome Sequencing

DNA extraction from cells was performed using the Promega Maxwell RSC PureFood GMO Kit (Promega Corporation, Madison, WI, USA). Genome sequencing was performed with the GridION X5 (Oxford Nanopore Technologies, Oxford, UK) followed by preparation of the genome library using a Rapid Barcoding Sequencing Kit (SQK-RBK004) (Oxford Nanopore Technologies, Oxford, UK). Read sequences were assembled using Unicycler (version 0.4.8) [26].

### 2.7. Genome Analysis

A genome sequence was annotated using DFAST (version 1.1.6, https://dfast.ddbj.nig.ac.jp/) [27]. AntiSMASH (version 5.0, https://antismash.secondarymetabolites.org/) [28]

was used for prediction of secondary metabolite gene clusters. A 16S rDNA sequence on the genome was analyzed to identify the strain using EzBioCloud (https://www.ezbiocloud.net/) [29]. Phylogenetic relationships were analyzed on the basis of 16S rDNA sequences using MEGA X (version 10.2.6) [30].

## 3. Results

### 3.1. Antagonistic Activities of Raw and Filtrate ADSs In Vitro

The five raw ADSs (AD, AF, AS, APF, and ASNF) significantly suppressed mycelial growth when compared with the control (Table 2). Of these, AD, AS, AF, and ASNF produced a clear inhibition zone. In contrast to the raw ADSs, the filtrate ADSs did not suppress mycelial growth (Table 2).

**Table 2.** Antifungal activity of raw and filtrate anaerobically digested slurries (ADSs) against Fos, indicated as mycelial growth by a co-culture test.

| Sample * | Mycelial Growth (mm) | | | | |
|---|---|---|---|---|---|
| | Raw | | Filtrate | | |
| AD | 56.1 | b | 85.6 | NS | |
| AS | 55.9 | b | 85.2 | | |
| AF | 60.8 | b | 85.5 | | |
| APF | 57.6 | b | 86.3 | | |
| ASNF | 53.5 | b | 86.4 | | |
| Control | 82.9 | a | 83.3 | | |

* ADSs generated from different source materials. AD, dairy cow manure; AS, sewage sludge; AF, food garbage; APF, pig manure + food garbage; ASNF, sewage sludge + night soil sludge + food garbage. The same letter indicates no significant difference based on Tukey's HSD ($p < 0.05$; $n = 4$). NS indicates not significant.

### 3.2. Antagonistic Activities of Bacterial Isolates from ADSs In Vitro

Overall, 32 strains were isolated from the five ADSs. Nine isolates were obtained from AD, and AD-3, AD-6, and AD-8 significantly suppressed the growth of Fos (Figure 1). Seven isolates were obtained from AF, and AF-1, AF-3, and AF-5 significantly suppressed the growth of Fos (Figure 1). Six isolates each were obtained from APF and ASNF, and APF-1 and ASNF-4 significantly suppressed the growth of Fos (Figure 1). Although four isolates were obtained from AS, no isolates suppressed the growth of Fos (Figure 1).

### 3.3. Effects of AD-3 on Fusarium Wilt of Spinach

The pot experiments were repeated twice (Exp 1 and Exp 2) and showed similar results. Disease severity with F+AD-3 was significantly ($p < 0.05$) lower than that with F (Table 3). The inhibitory effect of F+AD-3, as a percentage reduction in disease severity compared with that of F, was 64.3% (Exp 1) and 44.1% (Exp 2). As for Fos density in soil, there were no significant differences between F and F+AD-3 in either experiment (Figure 2). Fos density during cultivation was in the range of 4.3–4.8 CFU·g$^{-1}$ dry soil (Exp 1) and 3.9–4.4 CFU·g$^{-1}$ dry soil (Exp 2).

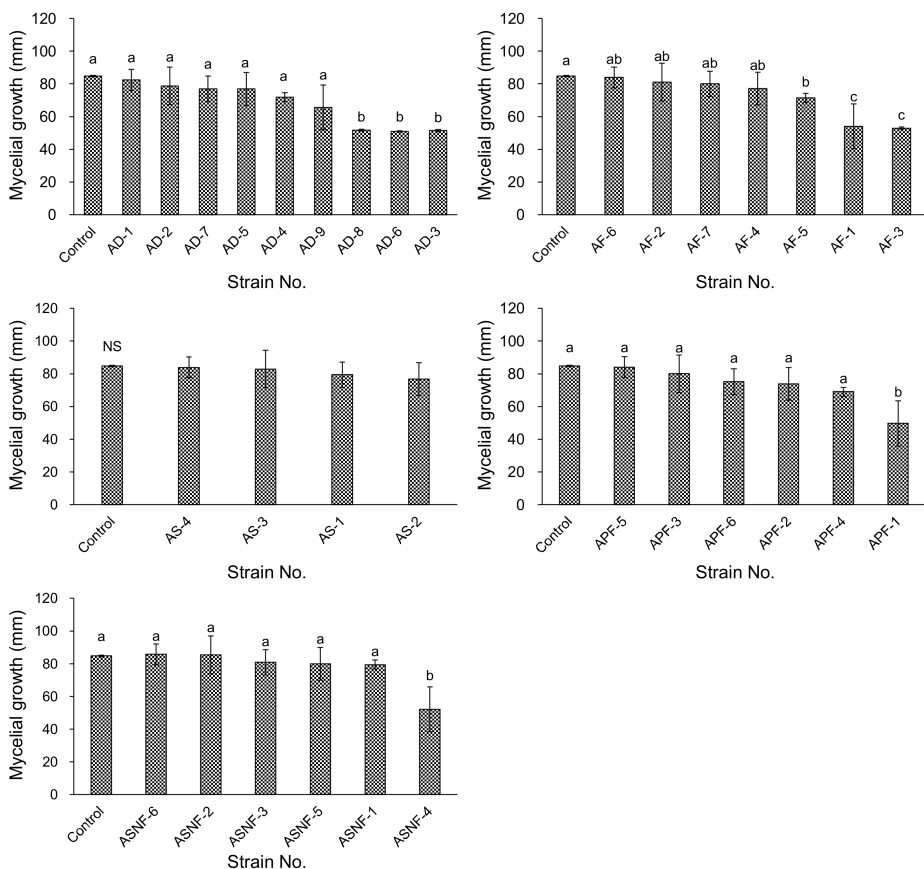

**Figure 1.** Antifungal activity of bacterial isolates from five anaerobically digested slurries (ADSs) against *Fusarium oxysporum* f. sp. *spinaciae* (Fos), indicated as mycelial growth (bars ± SD) by a co-culture test. AD, dairy cow manure; AS, sewage sludge; AF, food garbage; APF, pig manure + food garbage; ASNF, sewage sludge + night soil sludge + food garbage. SD, standard deviation of the mean. Bars with the same letter are not significantly different based on Tukey's HSD test ($p < 0.05$; $n = 4$).

**Table 3.** Disease severity (DS) of Fusarium wilt of spinach and inhibitory effect (IE) 28 d after seedling.

| | Treatment | | | | | | IE (%) [2] |
|---|---|---|---|---|---|---|---|
| | F | | F+AD-3 | | Untreated | | |
| Exp 1 [1] | 77.8 | a | 27.8 | b | 0.0 | c | 64.3 |
| Exp 2 | 95.8 | a | 53.6 | b | 0.0 | c | 44.1 |

[1] The pot experiment was repeated twice (Exp 1 and Exp 2). Values followed by the same letter within a row are not significantly different according to Tukey's HSD test [$p < 0.05$; $n = 9$ (Exp 1) and $n = 7$ (Exp 2)]. [2] IE, inhibitory effect.

### 3.4. Identification of AD-3

The morphological analysis showed that the strain AD-3 was a Gram-positive, rod-shaped bacterium. Based on the 16S rRNA gene sequence analysis, the closest species to strain AD-3 was *Bacillus velezensis*, showing 99.8% similarity. Additionally, phylogenetic analysis of the 16S rDNA indicated that strain AD-3 was positioned in the same group as other *B. velezensis* strains (Figure 3).

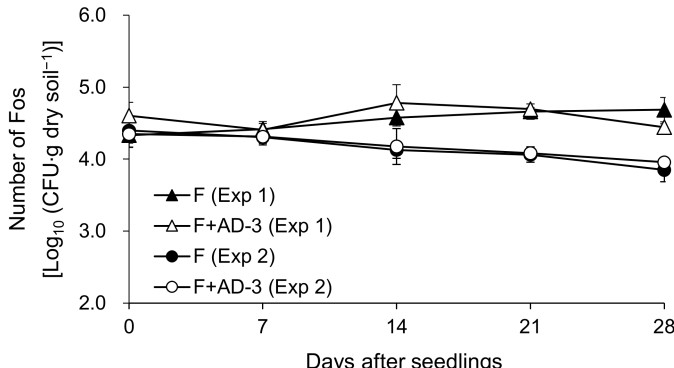

**Figure 2.** Density of *Fusarium oxysporum* f. sp. *spinaciae* (Fos) (plots ± SD) in soil after seedling emergence. SD, standard deviation of the means. Each plot represents the average of three replicates (pots). The experiment was repeated twice (Exp 1 and Exp 2).

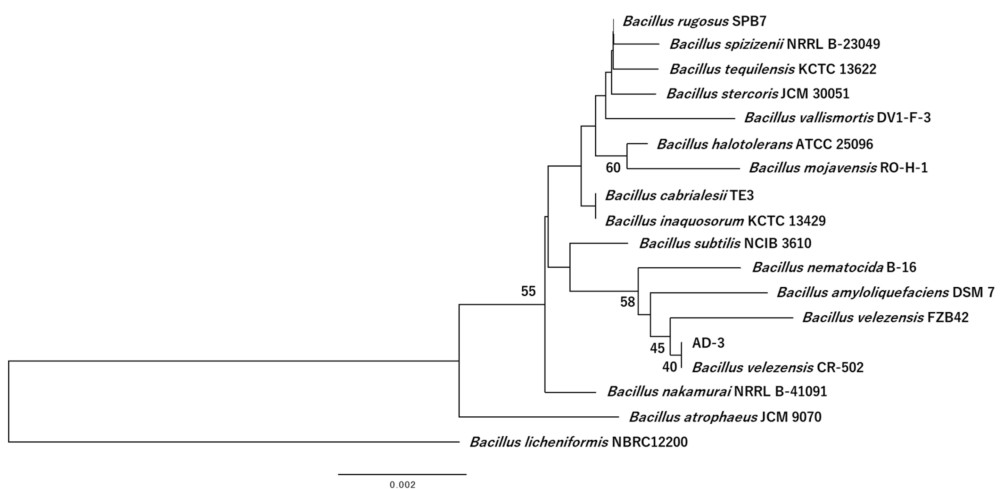

**Figure 3.** Phylogenetic tree based on 16S rDNA gene sequences of AD-3 strain and related species using neighbor-joining analysis. Bacillus licheniformis NBRC12200 served as an outgroup. Scale bar refers to a phylogenetic distance of 0.002 nucleotide substitutions per site. Bootstrap values were obtained based on 1000 replications.

*3.5. Genomic Characterization of AD-3*

The chromosomal and plasmid DNA of strain AD-3 were completely sequenced and annotated. Their sizes were 4.1 Mb and 220 Kb, respectively. Their INSDC (DDBJ/ENA/GenBank) accession numbers are AP024501 and AP024502, respectively. On the chromosomal DNA, genes related to the production of two kinds of non-ribosomal lipopeptides, fengycin/plipastatin (*pps* genes), and surfactin (*srf* genes), were predicted.

## 4. Discussion

The effect of the strain AD-3, isolated from AD, on Fusarium wilt disease incidence was assessed. In the present study, approximately $1.0 \times 10^4$ CFU·g$^{-1}$ dry soil was made and inoculated with AD-3 to achieve a concentration of $1.0 \times 10^6$ CFU·g$^{-1}$ dry soil. The pathogen density of the infected soil was mostly consistent with previous studies examining the effect of antagonistic bacteria against Fusarium diseases [31,32]. Inoculation of AD-3 into Fos-infected soil effectively reduced disease severity (by 64.3% and 44.1% in the two experiments). Thus, strain AD-3 effectively suppressed Fusarium wilt of spinach.

Strain AD-3 belongs to the *B. velezensis* group. *Bacillus velezensis* was recently reclassified as a synonym of several species including *Bacillus amyloliquefaciens* subsp. *plantarum*, *Bacillus methylotrophicus*, and *Bacillus oryzicola* [33,34]. *Bacillus velezensis* is frequently isolated from soil [35,36], rivers [37], and fermented food [38], and is recognized as a safe

biological resource in the field of biotechnology [38]. In the present study, we isolated AD-3 from ADS sourced from dairy manure. Many strains of *B. velezensis* exhibit biocontrol effects against plant pathogens and have been used to control common diseases of tomato, cucumber, lettuce, and wheat [39–42]. Our results revealed that AD-3 had the ability to control spinach Fusarium wilt. *Bacillus velezensis* FZB42, which is close to the strain AD-3, has gene clusters associated with the synthesis of secondary metabolites with antimicrobial activity [34,43,44]. The strain AD-3 had a gene cluster involved in synthesis of surfactin. This result corresponds with a part of Palazzini et al. (2016) reporting that the *Bacillus velezensis* strain possessed the gene cluster for several compounds including surfactin [45]. Surfactin is an important lipopeptide in the suppression of plant disease [46]. Yokota et al. (2015) reported that, although lipopeptides produced from *B. subtilis* suppressed Fusarium yellows, the reduction in pathogen density was slight [47]. Similar results were obtained in the present study; inoculation with AD-3 significantly suppressed Fusarium wilt of spinach without a reduction in pathogen density.

Five kinds of ADSs generated from different source materials suppressed the growth of Fos in vitro. There are several reports of factors that affect the suppression of plant pathogens. For example, Amari et al. (2000) reported that confrontation assay in vitro showed that ammonia and acetic acid in the slurry were the main factors influencing disease suppression [22]. Tao et al. (2012) reported that the supernatant of centrifuged ADS had a lower suppressive effect than raw ADS [21], and our results support this finding: filtrate ADSs did not suppress Fos growth in vitro (Table 2). Several antagonistic bacteria were isolated from ADSs (AD, AF, APF, and ASNF), except AS. Based on the results, the presence of bacteria is important for the inhibitory effect of ADSs.

The application of antagonistic microbes with organic amendment could provide more effective plant disease control than the use of a single strain alone [48,49]. In the present study, we applied only the strain AD-3 and assessed its suppressive effect on Fusarium wilt of spinach. To utilize ADSs in crop production, further study is needed to assess the effects and dynamics of AD-3 when AD is applied to infected soil.

**Author Contributions:** Conceptualization, T.S. and M.I.; methodology, T.S., K.T.N., N.T., Y.S. and M.I.; validation, T.S.; investigation, T.S., N.T. and Y.S.; data curation, T.S. and M.I.; writing—original draft preparation, T.S.; writing—review and editing, N.T. and M.I.; supervision, K.T.N. and M.I.; project administration, M.I.; funding acquisition, M.I. All authors have read and agreed to the published version of the manuscript.

**Funding:** This research was supported by grants from the Japan Association for Livestock New Technology.

**Data Availability Statement:** Not applicable.

**Conflicts of Interest:** The authors declare no conflict of interest. The funders had no role in the design of the study; in the collection, analyses, or interpretation of data; in the writing of the manuscript, or in the decision to publish the results.

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
