# Peer review of "Antagonism of Bacillus velezensis Isolate from Anaerobically Digested Dairy Slurry against Fusarium Wilt of Spinach"

_agronomy, doi:10.3390/agronomy12051058_

Round 1

Reviewer 1 Report

Line 149-152. Authors displayed the analysis using Tukey’s HSD test, and authors should provide more details, such as one-way analysis of variance.

Author Response

Dear Reviewer,

Thank you for reviewing my manuscript.

Here I answer to your comment.

Reviewer #1:

Line 149-152. Authors displayed the analysis using Tukey’s HSD test, and authors should provide more details, such as one-way analysis of variance.

> Accordingly, we re-wrote the statistical analysis in section 2.5.3 (lines 163-168).

Reviewer 2 Report

I understand that the manuscript describes the antagonistic activity of a B. velezensis isolate against F.o f.sp. spinaciae under laboratory conditions.

CRITICAL POINTS

1-The design of the plant-pathogen experiment and its interpretation. I do not see enough elements in the manuscript to ensure that there is protection due to AD-3 (see details below).
2-The analysis and discussion of bacterial biodiversity in the manuscript leaves me with doubts about the potential for agronomic use highlighted by the authors in lines 51/52.

DETAILS

TITLE
1-The title should be more specific. If the AD-3 genome has been sequenced, why not put the identified species in the title.
2-Working with soil pathogens, the term "suppressive" refers to soils whose microbial community reduces host infection, in this case I see more descriptive use of the term "antagonism" since it is what is obtained with a laboratory test.

MATERIALS AND METHODS
3-Some details must be specified in 2.2 (the diameter of the plates) and in section 2.4 (explain the procedures to adjust the pH of the soil and the final rates of NPK). It strikes me that with such care in the experiments the substrate was not sterilized.

4-In section 2.5.3, indicate the variety of spinach and the level of resistance (previously reported) it has to wilting by FOS. Differences in susceptibility to Fusarium wilt of cultivars are key to detecting differences in wilt. Seeds should be previously treated for control of Rhizoctonia, Pythium or other soilborne pathogens, to avoid confounding symptoms from these pathogens with Fusarium wilt assessments. If these procedures had not been performed previously, it was expected that reisolations would be done at the end of the experiment to be sure that the symptoms observed corresponded to FOS, especially if the soil and seeds were not treated.

RESULTS
5- Table 3 is expected that the severity is indicated according to the observations of the scale of degrees 0-5 (line 143/ 145), not in %. Values ​​of the dispersion of the mean should be given, that would explain that there are no significant differences between 27.8 and 53.6. for example. I would expect to be able to identify the n by treatments. This table should compare not only the differences between experiments, but also between control and treated.
6- Figure 2 is better visualized if only the amplitude of the Y axis is reduced to the range of values ​​of interest.

DISCUSSION

7- Strictly speaking, AD-3 was not introduced into infested soil (infected refers to the plant with the pathogen inside), but both microorganisms were applied concomitantly. Again, the experiment was evaluated in degrees of a scale, not in percent (line 232).

8-In lines 249/250 it is stated that "AD-3 significantly 249 suppressed Fusarium wilt", however the results shown in table 3 only compare experiments 1 and 2 (in column), not the treatments.

9- In lines 250/251 it is stated that AD-3 produces surfactin, which is very speculative with the data shown since the experiments do not include experiments with pure surfactin or with fractions extracted from crude filtrates. That AD-3 have the genes does not mean that they express it under the conditions tested in these experiments.

10- If the amount of inoculum in the soil is determined, it is expected that these results will be discussed. Likewise, an explanation was expected for the fact that the ADSs filtrates had no effect, and raw materials did.

11- What is stated in lines 260/261 about the combined use of bacterial isolates is correct, however, in this study of six isolates with a significant inhibitory effect with respect to the control, only AD-3 was used. It is inexplicable to me why the opportunity to investigate the rest of the strains was wasted.

Author Response

Dear reviewer,

Thank you for reviewing my manuscript.

Here I answer to your comments.

Reviewer #2:

1-The title should be more specific. If the AD-3 genome has been sequenced, why not put the identified species in the title.

2-Working with soil pathogens, the term "suppressive" refers to soils whose microbial community reduces host infection, in this case I see more descriptive use of the term "antagonism" since it is what is obtained with a laboratory test.

> We agree your suggestion. Accordingly, we changed the title as follows:

“Antagonism of Bacillus velezensis isolate from anaerobically digested dairy slurry against Fusarium wilt of spinach”.

3-Some details must be specified in 2.2 (the diameter of the plates) and in section 2.4 (explain the procedures to adjust the pH of the soil and the final rates of NPK). It strikes me that with such care in the experiments the substrate was not sterilized.

> We added the diameter of the plates in 2.2 (lines 84/85) Methods for adjusting soil pH and fertilizer components were added in 2.5.3 (lines 131-135).

4- In section 2.5.3, indicate the variety of spinach and the level of resistance (previously reported) it has to wilting by FOS. Differences in susceptibility to Fusarium wilt of cultivars are key to detecting differences in wilt. Seeds should be previously treated for control of Rhizoctonia, Pythium or other soilborne pathogens, to avoid confounding symptoms from these pathogens with Fusarium wilt assessments. If these procedures had not been performed previously, it was expected that reisolations would be done at the end of the experiment to be sure that the symptoms observed corresponded to FOS, especially if the soil and seeds were not treated.

> We added a note about cultivar and its susceptibility to Fusarium wilt in section 2.5.3 (lines 143/144). We selected the spinach cultivar reported to susceptible to Fusarium wilt. We confirmed that Fos was detected from wilted spinach at least in Exp 1 to ascertain if the symptoms of wilting are due to Fos.

5- Table 3 is expected that the severity is indicated according to the observations of the scale of degrees 0-5 (line 143/ 145), not in %. Values of the dispersion of the mean should be given, that would explain that there are no significant differences between 27.8 and 53.6. for example. I would expect to be able to identify the n by treatments. This table should compare not only the differences between experiments, but also between control and treated.

> We agree the point that the values in Table 3 is the scale, not in %. To clearly indicate the effect of AD-3, the inhibitory effect as a percentage reduction in disease severity was added in section 2.5.3 (lines 159/160). The n by treatments also added in section 2.5.3 (lines 161/162).

6- Figure 2 is better visualized if only the amplitude of the Y axis is reduced to the range of values of interest.

> We reduced the amplitude of the Y axis (Figure 2).

7- Strictly speaking, AD-3 was not introduced into infested soil (infected refers to the plant with the pathogen inside), but both microorganisms were applied concomitantly. Again, the experiment was evaluated in degrees of a scale, not in percent (line 232).

> We added the inhibitory effect (%) as a percentage reduction in disease severity to evaluate in percent in lines 159/160, 216.

8-In lines 249/250 it is stated that "AD-3 significantly 249 suppressed Fusarium wilt", however the results shown in table 3 only compare experiments 1 and 2 (in column), not the treatments.

> Sorry for misrepresentation. We re-wrote the caption of Table 3. We compared the treatment (in row), not experiments (in column). Some details for statistical analysis was added in section 2.5.3.

9- In lines 250/251 it is stated that AD-3 produces surfactin, which is very speculative with the data shown since the experiments do not include experiments with pure surfactin or with fractions extracted from crude filtrates. That AD-3 have the genes does not mean that they express it under the conditions tested in these experiments.

> We did not do experiment with pure surfactin. Accordingly, we deleted the text in lines 250/251 “The strain AD-3 produced the secondary metabolite surfactin and thus suppressed the disease.”.

10- If the amount of inoculum in the soil is determined, it is expected that these results will be discussed. Likewise, an explanation was expected for the fact that the ADSs filtrates had no effect, and raw materials did.

> We added a discussion related to the amount of inoculum in the soil (lines 254-258).

11- What is stated in lines 260/261 about the combined use of bacterial isolates is correct, however, in this study of six isolates with a significant inhibitory effect with respect to the control, only AD-3 was used. It is inexplicable to me why the opportunity to investigate the rest of the strains was wasted.

> We added the note about the reason for using AD-3 in 2.5.2 (lines121/122) as follows:

In 2.4, AD had several effective bacteria against Fos among ADSs. Of the several bacteria in AD, AD-3 which suppressed Fos growth the most was used for pot experiment.

Reviewer 3 Report

The manuscript entitled Suppressive effect of anaerobically digested slurries and 2 bacterial isolates against Fusarium wilt of spinach is an interesting article with scientific importance. 

-authors needs to add the most recent references into the introduction with more desciption

-materials and methods should be complete somewhere it looks short

-authors needs to check the phylogenetic tree I am not satisfying in present form

-discussion looks very short

-conclusion section can be improve

Author Response

Dear reviewer,

Thank you for reviewing my manuscript. Here I answer to your comments.

Reviewer #3

-authors needs to add the most recent references into the introduction with more description

> We replaced some references with more recent references.

-materials and methods should be complete somewhere it looks short

> We added some details for the procedure in section 2.5.3.

-authors needs to check the phylogenetic tree I am not satisfying in present form

> We reconstructed a phylogenetic tree by adding sequences of several strains close to AD-3 (Figure 3).

-discussion looks very short

> We added a note about a discussion related to experimental condition, efficacy of AD-3.

-conclusion section can be improve

> We improved the conclusion in section of discussion (lines 257/258, 286).

Again, thank you for giving us the opportunity to strengthen our manuscript with your valuable comments and queries.